# Microbiota alters the metabolome in an age- and sex- dependent manner in mice

Kirsty Brown [1], Carolyn A. Thomson[1], Soren Wacker [2], Marija Drikic[2], Ryan Groves[2], Vina Fan [1], Ian A. Lewis [2] & Kathy D. McCoy [1] ✉

Commensal bacteria are major contributors to mammalian metabolism. We used liquid chromatography mass spectrometry to study the metabolomes of germ-free, gnotobiotic, and specific-pathogen-free mice, while also evaluating the influence of age and sex on metabolite profiles. Microbiota modified the metabolome of all body sites and accounted for the highest proportion of variation within the gastrointestinal tract. Microbiota and age explained similar amounts of variation the metabolome of urine, serum, and peritoneal fluid, while age was the primary driver of variation in the liver and spleen. Although sex explained the least amount of variation at all sites, it had a significant impact on all sites except the ileum. Collectively, these data illustrate the interplay between microbiota, age, and sex in the metabolic phenotypes of diverse body sites. This provides a framework for interpreting complex metabolic phenotypes and will help guide future studies into the role that the microbiome plays in disease.

In mammals, all nutrients are introduced through the gastrointestinal tract (GIT) and absorbed across the intestinal epithelium. Processing this stream of nutrients into the molecular building blocks of life is a fundamental biological function that affects virtually all aspects of our health[1]. Although mammalian metabolism has been intensively studied for over a century[2,3], we know relatively little about how microbes living within the GIT shape the complement of metabolites that ultimately become available to the host. Microbes have diverse metabolic capacities[4]; thus changes in the composition of the microbiome can have a direct impact on the metabolic capacities within the GIT. Some of these perturbations are known to affect biologically relevant molecules, such as the bioavailability of vitamins and fermentation of proteins and carbohydrates[5–7]. Consequently, changes in the microbiome composition can affect which molecules are present within the GIT, transit the epithelial barrier, and become available to host cells.

A growing body of literature has shown that changes in the composition of the microbiome can have profound impacts on immune function, neuromodulation, and disease progression. Metabolites derived from intestinal bacteria penetrate to systemic host tissues[8] and are one mechanism through which bacteria can influence host physiology. To date, much focus has been placed on short-chain fatty acids

(SCFA), bacterially derived breakdown products of undigestible complex carbohydrates and proteins[9], due to their pleiotropic roles in immune and metabolism regulation. Several other microbe-derived metabolites have immune-regulating function, including tryptophan metabolites[10], lactate[11], inosine[12], and bile acids[13]. However, with recent advances in our ability to assess metabolite profiles in vivo, we are starting to realize that this small selection of metabolites may represent only the tip of the iceberg when it comes to bacterial metabolite-mediated immune regulation.

Modulation of immunity is only one axis whereby microbiome-induced metabolites can affect host health. The GIT hosts the enteric nervous system, a major branch of the peripheral nervous system, and has direct connections to major metabolic organs, such as the liver and pancreas. As such, the intestinal microbiome can influence metabolites that act as signaling molecules within the nervous system[14] or participate directly in the metabolic capacity of the host through extraction of nutrients or stimulation of host hormones[15].

Although we know that many of these complex host-microbiome dynamics are modulated through metabolism, the overlapping effects of host and microbial metabolism makes it challenging to identify microbe-induced metabolites that contribute to health or disease.

[1]Department of Physiology and Pharmacology, Snyder Institute of Chronic Diseases, Cumming School of Medicine, University of Calgary, Calgary T2N 4N1, Canada. [2]Department of Biological Sciences, University of Calgary, Calgary T2N 1N4, Canada. ✉e-mail: kathy.mccoy@ucalgary.ca

Recently, investigators have started using germ-free (GF) mouse models to better understand these host-microbe-metabolite dynamics, including a recent study that mapped mouse metabolism in adult female GF and SPF mice and found that the microbiota affects the chemistry of all organs[16]. Earlier studies investigated the metabolome of germ-free mice to a limited degree[17,18]. However, many previous studies were restricted with respect to the number of body sites studied[18], or by analytical methods that cannot fully identify metabolites[16,18]. Moreover, little is known about the interplay of other biological factors, such as age and sex, on these microbially-induced metabolic changes.

Here, we leveraged gnotobiotic mouse models and liquid chromatography-mass spectrometry (LC-MS) to systematically characterize metabolic profiles throughout the length of the GIT at different stages of development in germ free (GF) and specific pathogen-free (SPF) mice, as well as gnotobiotic mice colonized with the Oligo-MM12 (OMM12) consortia. The latter is composed of 12 murine commensal bacterial species that represent the five major prokaryotic phyla in the murine GIT (Bacteroidota, Bacillota, Pseudomonadota, Actinomycetota and Verrucomicrobiota[19]). Complementing data generated from the GIT, we profiled metabolites within the peritoneal fluid, serum, liver, spleen, and urine to understand the impact of colonization on systemic sites. We include data from weanlings, adult mice, and equal numbers of male and female mice, to determine the relative contribution of microbiota, age, and sex to metabolome, as well as interactions between these factors. We report on changes in metabolite levels – assigned by accurate mass and co-retention with metabolite standards – across body sites, ages, sex, and hygiene status. These data allow us to map the interplay of the microbiome with key biological variables and acts as a resource upon which future research can be built.

## Results

### Anatomically and functionally distinct tissues have unique metabolic signatures

To gain a broad understanding of the factors driving host metabolism, we profiled metabolites in contents from different regions of the GIT and in systemic sites (peritoneal fluid, serum, liver, spleen, and urine) of differentially colonized male and female C57BL/6 mice at 3, 8 and 12 weeks of age. Mice were either GF, gnotobiotically colonized with OMM12, or SPF. When data were compared as a whole, we found that body site had the largest effect on the overall metabolome of a sample (Supplementary Table 1). However, microbiota, age and sex all had a significant effect, with sex explaining the least of the variation (Supplementary Table 1). We then used linear models on metabolite abundance to determine the proportion of variation that was attributable to each factor (site, microbiome, age, and sex) (Supplementary Fig. 1A). The levels of most metabolites differed mainly by site, as expected. However, a small number of specific metabolites were affected by microbiota, age, or sex (Supplementary Figure 1A).

The GIT is central to host metabolism and the environment along the tract is highly variable with regional specialization in terms of nutritional absorption, immune modulation, and microbial burden[20]. In keeping with this, we found that each site within the GIT had distinct metabolite signatures; however, the lower GIT metabolome of GF mice largely resembled the upper GIT tract of colonized mice, suggesting that this phenomenon was at least partially microbe-dependent (Fig. 1A, Supplementary Figure 2). Dietary amino acids, including tryptophan, leucine, and histidine were present in higher abundance in the upper GIT and decreased towards the lower GIT (Fig. 1B), likely because of absorption into the bloodstream. The relative abundance of sugars, nucleosides and fatty acids generally increased in the lower GIT, with microbially-induced changes apparent in the lower GIT illustrated by the higher abundance of sugars and lower abundance of fatty acids and nucleosides in GF mice (Fig. 1B). Therefore, although

anatomical location had the greatest impact on metabolism overall, metabolic differences within the GIT were partially attributable to microbial exposure, age, and sex.

### Relative contribution of microbiota, age, and sex to metabolism

To generate an overall view of how microbial regulation of metabolism is influenced by other biological factors, we analyzed the relative impact of microbiome, age, and sex on metabolism at sites within the GIT and systemically (Supplementary Table 2). Microbiota had a significant effect at all sites profiled (Fig. 2A) and explained the most metabolic variation at all sites within the GIT (Fig. 2A). In the serum, urine and peritoneal fluid, microbiota and age explain similar proportions of variation, while in the liver and spleen, age had a larger effect than microbiome (Fig. 2A). Sex differences explained the least amount of variation in all sites analyzed. In line with this, a high proportion of metabolites were altered by colonization in the cecum and colon, while age affected more metabolites in the liver and spleen (Fig. 2B). Using linear models, we mapped the proportional extent to which each metabolite was affected by age, microbiome, and sex within each site, generating an overview of the impact these factors had on individual metabolites (Fig. 2C & Supplementary Figure 1B).

### Metabolism is altered by microbial complexity

To understand how the density and complexity of the bacterial community contributes to metabolism, we compared the metabolome of GF, OMM12 and SPF colonized mice (Supplementary Fig. 2). As expected, colonization had the most pronounced impact on metabolites in the lower GIT, regardless of microbiota complexity, with OMM12 significantly altering the abundance of 49% and 54% of metabolites, and the SPF microbiota regulating 52% and 62% of metabolites, in the cecum and colon respectively (Fig. 3A). Since the cecum and colon are the most densely colonized regions, it is logical that they may be central to bacterial contributions to host metabolism. However, our data showed that colonization also had an impact on metabolism in the upper GIT and all other sites sampled (Fig. 3A).

The individual commensal species that make up the OMM12 community have unique metabolic profiles in vitro (Supplementary Figure 3A, B) and composition of the OMM12 consortia differs along the length of the GIT (Supplementary Figure 3C). To determine if observed metabolic phenotypes could be tied directly to microbial composition of each site, we compared the changing metabolic abundances and microbial composition at each site in the GIT (Supplementary 3D) to the to the unique metabolic phenotypes of each individual species from the OMM12 community. Although this analysis showed co-variance of microbial community composition and metabolic composition, the complex cross-feeding between organisms (as well as host/microbe dynamics) obscured the individual species phenotypes sufficiently to draw any firm conclusions with respect to which individual species were responsible for metabolic profiles observed in the GIT. However, our data clearly indicate that microbial community composition affects the GIT metabolism profile. We observed sets of molecules that were modulated by microbiome community composition (SPF microbiota, OMM12 microbiota, or modulated similarly in both relative to GF; Fig. 3A–C, Supplementary Figure 4). To better illustrate these differences, we selected metabolites that were significantly different in either OMM12 or SPF versus GF mice (Supplementary Figure 4A) and compared the fold change in the intensity of these metabolites in both colonizations (Fig. 3B, Supplementary Figure 4B). This analysis highlighted numerous metabolites that were influenced microbial community composition (Fig. 3B, Supplementary Figure 4B). For example, in the upper GIT, both OMM12 and SPF colonized mice had higher levels of cholate than GF mice, while taurolithocholate was increased only in SPF mice (Fig. 3C). In the lower GIT, allantoin was consumed and glutamine and guanine were produced

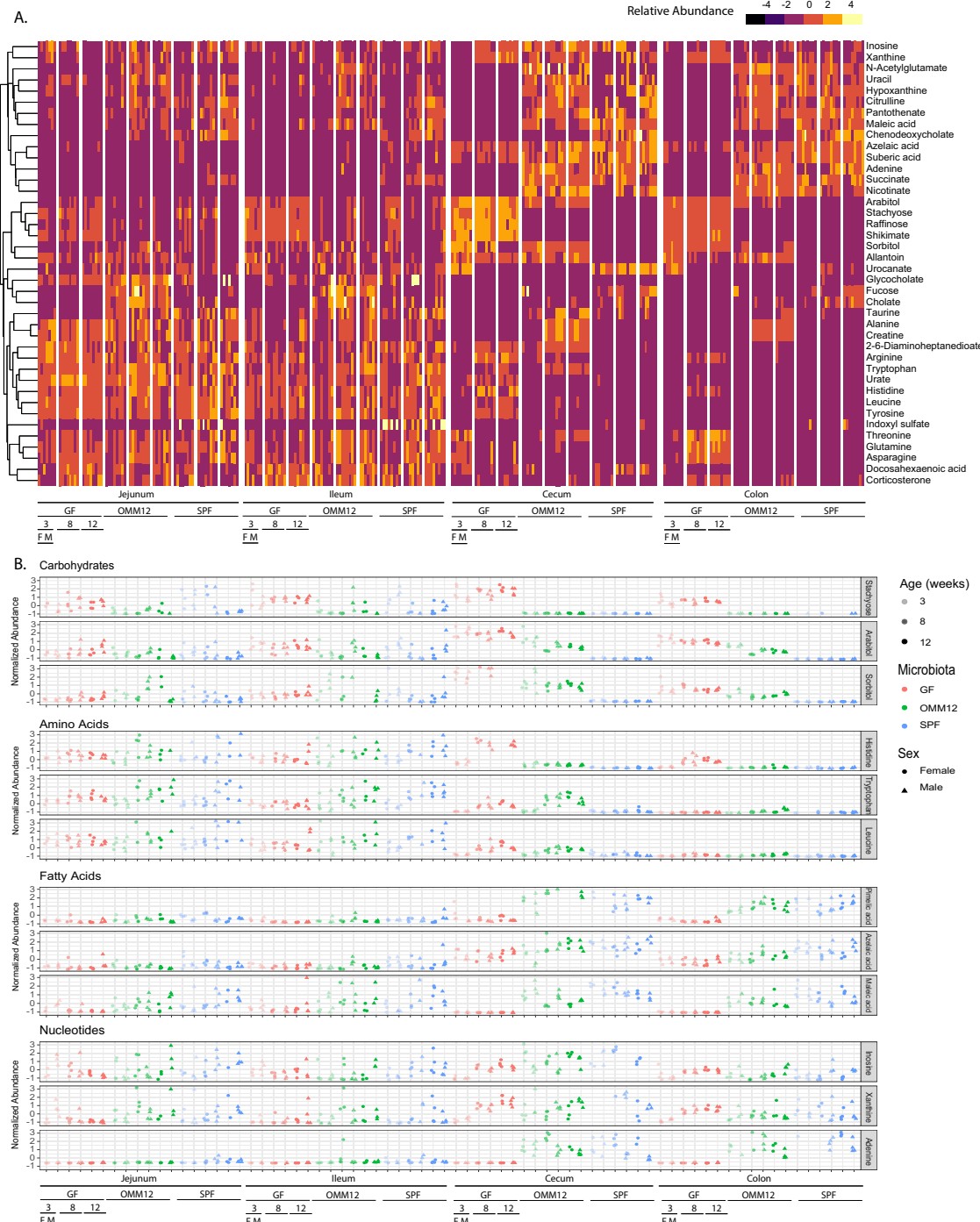

**Fig. 1 | Anatomical survey of intestinal tract metabolism. A** Heatmap depicting the 40 most variable metabolites in samples from luminal content of the jejunum, ileum, cecum, and colon. **B** Scatterplots showing the average relative abundance of carbohydrates, nucleosides, amino acids, and fatty acids that were differentially abundant along the GIT. Samples are ordered as indicated with a total of $n = 72$ samples/site (equal representation from male and female mice, GF, OMM12 and SPF colonized mice and 3-, 8- and 12-week-old mice). GF germ-free, OMM12 Oligo-MM12, SPF specific pathogen free, M male, F female. Source data are provided as a Source Data file.

only by the SPF microbiota, while raffinose was consumed and nicotinate was produced by both the OMM12 and SPF microbiota.

The microbial community-linked changes in metabolic profiles were also observed in a range of sites outside the GIT. For example, hippurate and indoxyl sulfate were increased only in SPF mice, whereas 3-hydroxybutyrate was decreased in the serum and spleen of SPF mice compared to GF mice (Fig. 3C). Together, our data highlight metabolic preferences of microbial communities, which has an impact on metabolite concentration within the GIT and systemically.

Our data suggests that some metabolites may be altered as a generic response to colonization whereas others may be dependent on bacterial complexity or the presence or absence of certain keystone taxa.

### Microbiota-induced metabolites that are age-dependent
Age has a considerable impact on metabolism (Fig. 2A), and as such, we sought to understand if age-related changes in metabolism were dependent on the microbiota. To this end, we determined which

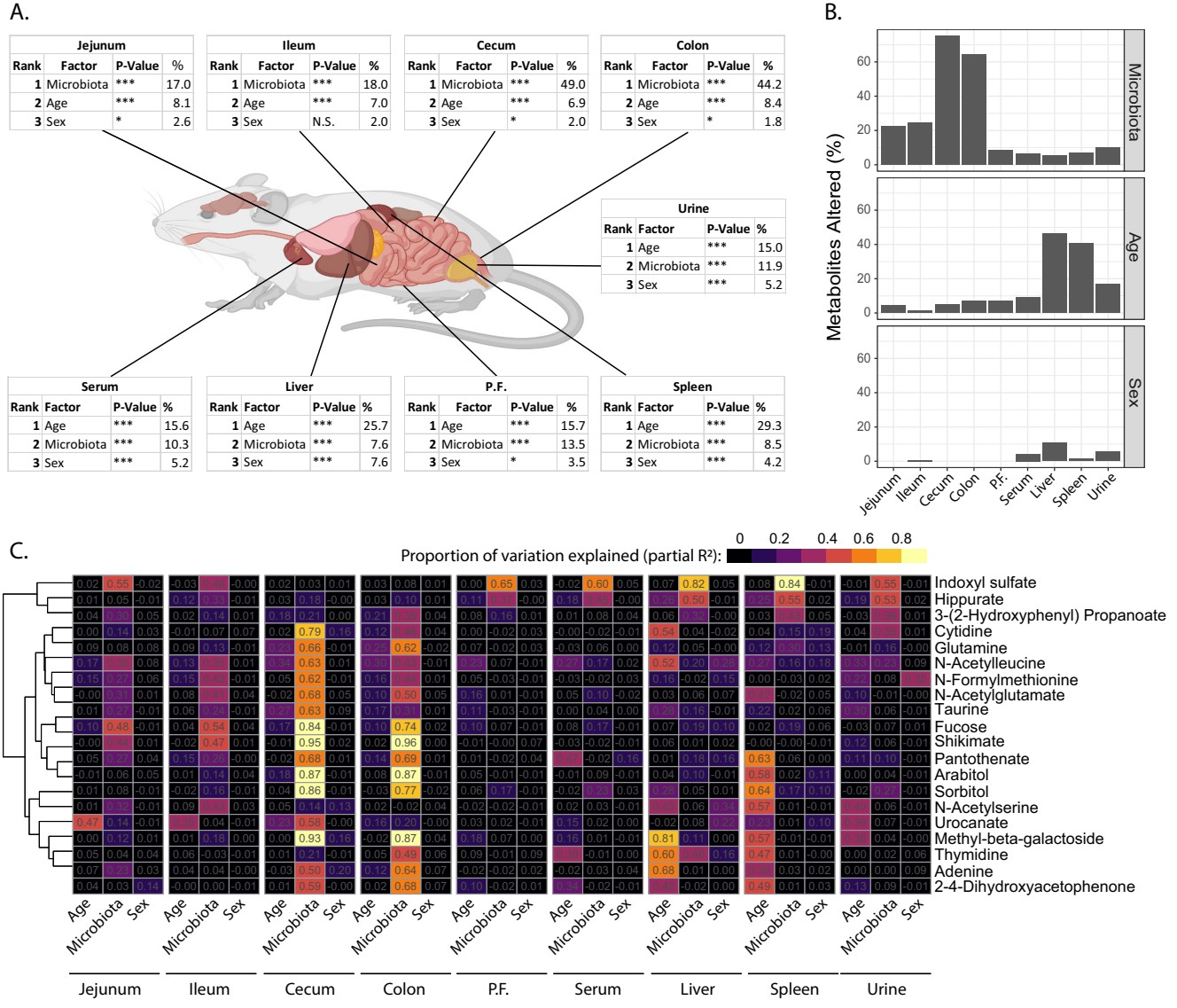

**Fig. 2 | Contribution of microbiome, age, and sex to metabolism. A** Estimated proportion of variation explained by microbiota, age and sex in each site sampled based on PERMANOVA statistics. **B** Proportion of metabolites altered (*P*Adj <0.05) by microbiome, age, and sex at each site. **C** Partial R² showing the relative contribution of age, microbiota, and sex in explaining the variation in metabolite abundance at each site. Metabolites shown are the 20 metabolites where the most variation was explained by the three factors, full heatmap for all metabolites is shown in Supplementary Figure 1. Data is representative of *n* = 72 samples / site (equal representation from male and female mice, GF, OMM12 and SPF colonized mice and 3-, 8- and 12-week-old mice). P.F. = peritoneal fluid. Parts of Fig. 2A were generated using Biorender.org under license. Source data are provided as a Source Data file.

metabolites were changed in young and adult mice under the three microbiota conditions at each sample site (Supplementary Figure 5A). We then selected metabolites that were significantly affected by age, in either GF or SPF mice, and compared the fold change in young versus adult GF and SPF mice (Fig. 4A, Supplementary Figure 5B). Although some metabolites were similarly affected by age in GF and SPF mice, particularly in systemic sites, many metabolites were affected by age in a microbiome-dependent manner (Fig. 4A, Supplementary Figure 5B, C). For example, lower GIT of 8-week-old GF mice had increased levels of histidine, uridine, glutamine, ornithine, and threonine (Fig. 4A, B) relative to 3-week-old GF mice. In contrast, most metabolites at systemic body sites were similarly affected in GF and SPF mice (Fig. 4A, B). Exceptions to this included increased 4-Acetamidobutanoate in the liver and peritoneal fluid of adult GF mice, which was not present at 3 weeks of age but increase in adulthood, as well as low levels of malate and succinate in the urine of young GF mice, which also increased with age (Fig. 4B). Together,

these results indicate that the microbiota can differentially affect metabolism at different developmental stages.

## Microbiota induces metabolites in a sex-dependent manner

Of the experimental variables tracked in this study, sex had the smallest overall influence over metabolic phenotypes. However, sex was still an important biological factor with significant sex-linked metabolic differences observed in all sites except the ileum (Fig. 2A). One important question we sought to understand is if sex plays a role in microbiome-induced metabolic phenotypes. The production of sex hormones increases with age, and when we compared sex-induced changes in metabolism at all sites, we observed more changes in metabolism in adult mice than weanlings (Supplementary Figure 6). As such, we focused our analyses on adult mice (8–12 weeks of age). To understand connections between microbial colonization and induction of metabolites in male and female mice, we selected metabolites that were differentially abundant based on sex in either GF or SPF mice.

**Fig. 3 | Impact of microbial composition on metabolism. A** Number of metabolites that are differentially abundant (PAdj <0.05) in OMM12 vs. GF (red), SPF vs. GF (green) and OMM12 & SPF vs. GF (blue) in each sample site. **B** Biplot showing the Log2FC in SPF vs. GF and OMM12 vs. GF in metabolites that were significantly different (PAdj <0.05) from GF in either colonization (red dots = OMM12, green dots = SPF) or both colonizations (blue dots). **C** Violin plots of highlighted metabolites from **B**. Data represents 24 mice per group with equal representation from male and female mice and mice that are 3-, 8-, and 12- weeks of age. GF germ-free, OMM12 = Oligo-MM12, SPF-specific pathogen free. Source data are provided as a Source Data file.

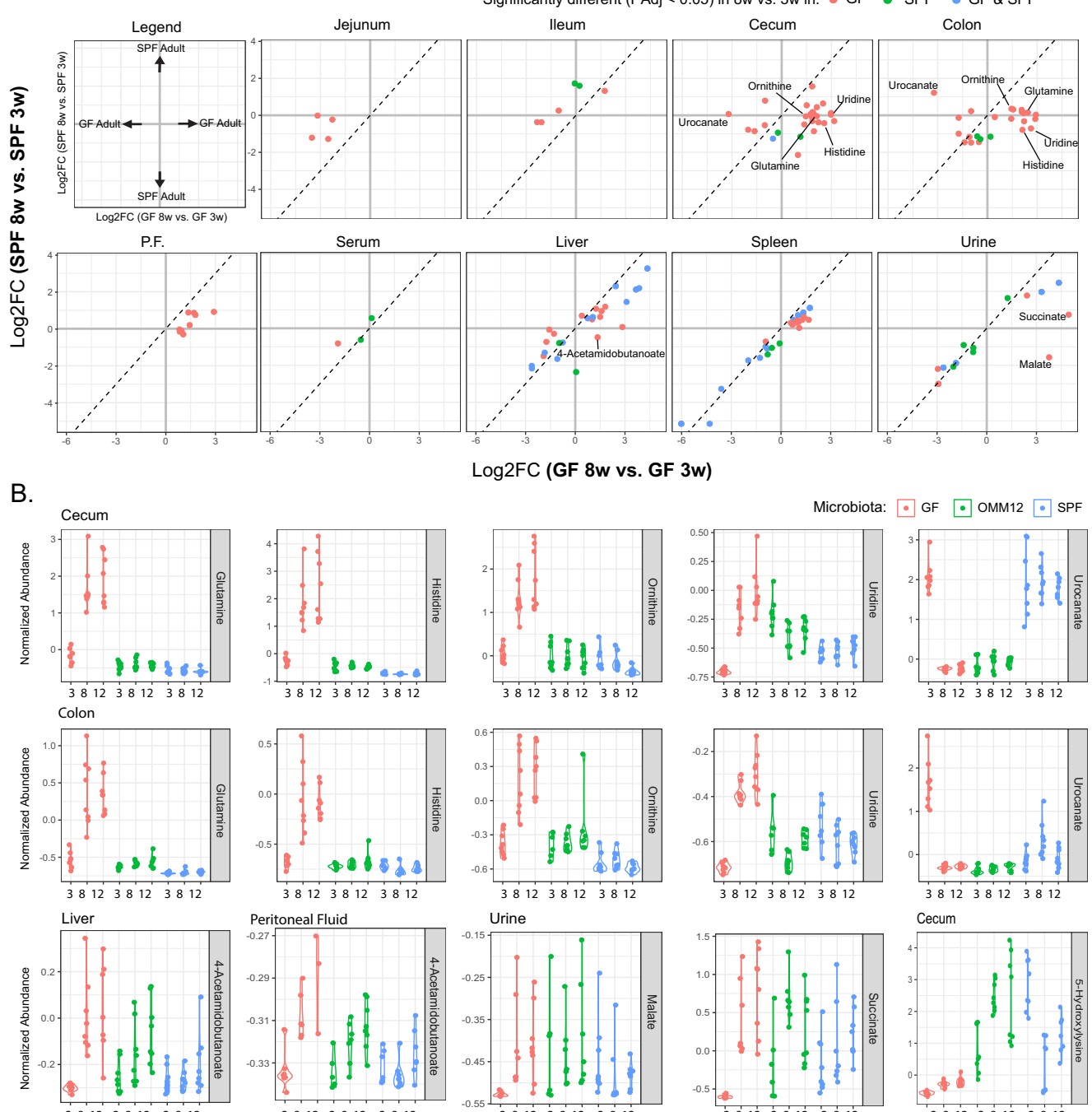

**Fig. 4 | Age-specific changes in microbiome-associated metabolism. A** Biplot showing the Log2FC in metabolites that were significantly different (PAdj <0.05) in 3-week-old vs. 8-week-old SPF mice or 3-week-old vs. 8-week-old GF mice. Metabolites were significantly different based on age in GF (red) or SPF (green) or both GF and SPF (blue). **B** Violin plots of highlighted metabolites from A that are differentially affected by colonization in young and adult mice. Data shown for GF, OMM12 and SPF colonized mice. Data represent 8 mice / group (n = 4 F and n = 4 M). P.F. peritoneal fluid. GF germ-free, OMM12 Oligo-MM12, SPF specific pathogen free, M male, F female. Source data are provided as a Source Data file.

Aside from some specific differences in the cecum, colon and liver, few of the annotated metabolites that were affected by sex were differentially altered in GF versus SPF mice (Fig. 5A, Supplementary Figure 7A, B).

Although many of the sex-dependent metabolites were similarly modulated in GF and SPF mice, there were several exceptions. Whereas GF mice showed no sex-linked differences in adenine, inosine, indoleacetic acid, or N-Acetylserine levels in the lower GIT, colonized mice

had significant sex-linked differences in these metabolites. Specifically, male SPF mice had higher levels of inosine and lower levels of adenine compared to female SPF mice and OMM12 had intermediate phenotypes (Fig. 5B). Indoleacetate was increased in response to SPF colonization in male but not female mice and N-Acetylserine was consumed only in male mice that were colonized with SPF microbiota. We also observed sex-specific changes in microbe-induced metabolism in the liver, whereby 5-Oxoproline was uniquely increased in male mice that

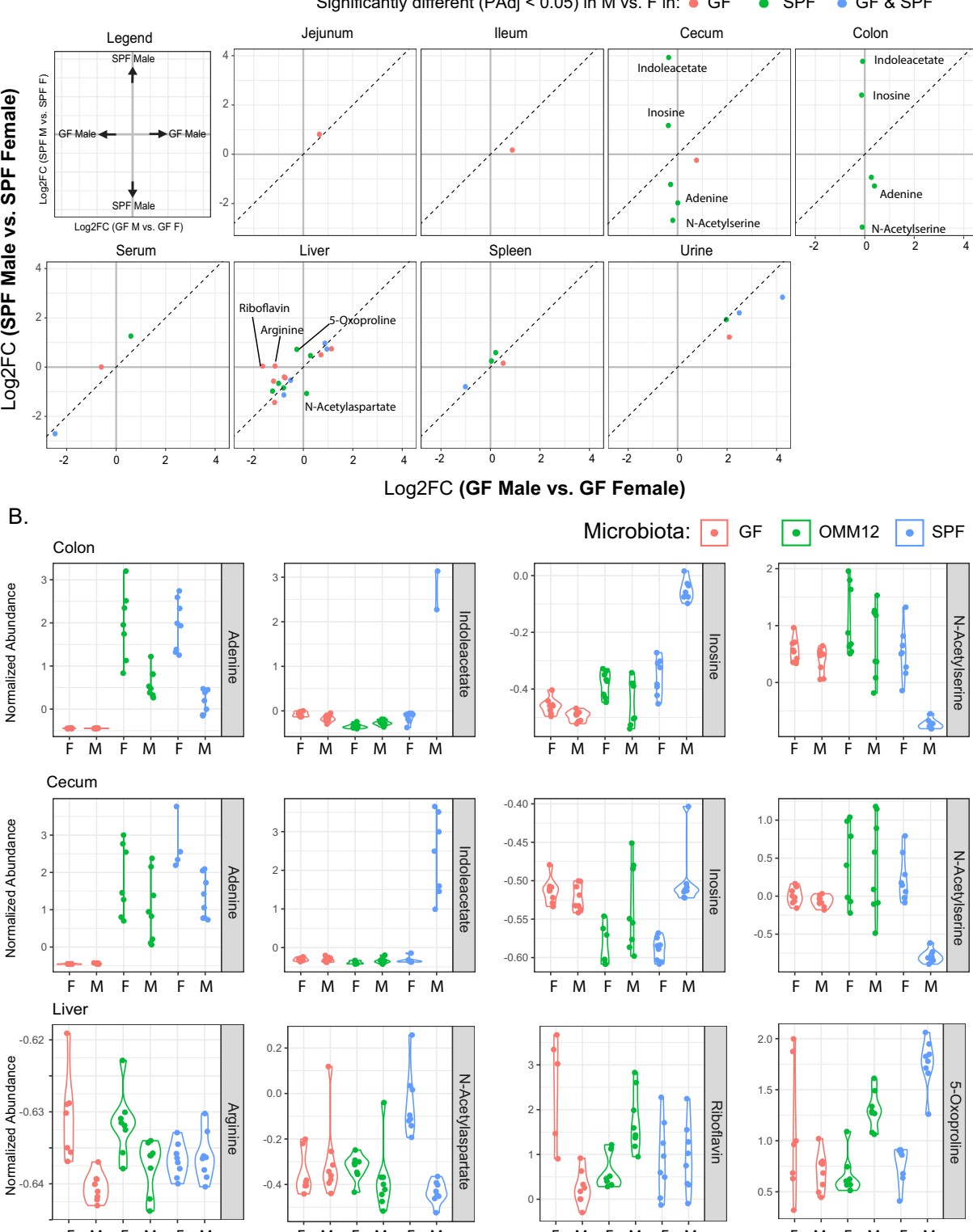

**Fig. 5 | Sex-specific changes in microbiota-associated metabolism. A** Biplot showing the Log2FC in metabolites that were significantly different (PAdj <0.05) in male vs. female SPF mice or male vs. female GF mice. Metabolites were significantly different based on age in either GF (red) or SPF (green) or in both GF and SPF (blue). **B** Violin plots of highlighted metabolites from A that are differentially affected by colonization in male and female mice. Data shown for GF, OMM12 and SPF colonized male and female mice. Data represented 8 mice/group. GF germ-free, OMM12 Oligo-MM12, SPF specific pathogen free, F female, M male. Source data are provided as a Source Data file.

were colonized with either SPF or OMM12 and N-Acetylaspartate was elevated only in SPF-colonized female mice. Riboflavin and arginine were increased in the liver of female mice that were GF, but this phenotype was not observed in colonized mice. Taken together, these data show that the microbiome can play a pivotal role in eliciting sex-based metabolic differences between mice.

## Discussion

The interplay between diet, microbiome and metabolites is an exciting field of research, and it is not surprising that large-scale clinical studies have begun to draw connections between these three components and the impact they have on host physiology[21]. Including the metabolome as a functional aspect of host physiology has allowed for studies that connect host and microbial transcriptional signatures to functional aspects of disease progression[22]. Leveraging microbe-metabolite-host interactions towards personalized medicine holds great promise for improving health and preventing disease.

Due to its constant exposure to dietary and microbial compounds, the mucosal surface of the GIT is central to host metabolism. Moreover, as it interacts with key metabolic organs and hosts substantial components of the body's immune and nervous systems, the GIT serves as a crucial interface between host, diet, and microbiota. In this study, we have systematically characterized microbial regulation of metabolism in the context of age and sex. We quantify the relative impact that microbiota, age and sex have on steady-state metabolite levels and describe cases where microbial-induced metabolism is affected by the age and sex of the animal (Supplementary Figure 8: Summary Figure). Our work sheds light on microbe-metabolite-host interactions and provides a resource that will aid future research into host-microbe-metabolite interaction.

The anatomy and physiology of the GIT exhibits regional specialization. Along the length of the GIT there are changes in immune, epithelial, and stromal cell populations, concurrent with changes in mucus structure, pH, oxygen availability and secretion of antimicrobial compounds[20]. Within the GIT, the metabolome was highly dependent on site with a decrease in dietary compounds such as amino acids and carbohydrates and an increase in compounds that are produced by bacteria in the lower GIT. The high proportion of metabolites that are altered within the lower GIT of colonized mice speaks to the massive contribution that microbiota has on the metabolites that become available for utilization by the host. Large-scale clinical studies have begun to correlate the abundance of bacterial taxa in feces with the concentration of serum metabolites[23], highlighting the interconnectedness of lower GIT bacterial metabolism and host metabolic phenotypes.

The unique metabolic capacity of different microbiotas is highlighted in our study by comparing the metabolome of SPF and OMM12 colonized mice to GF mice. We found that the commensal species in the OMM12 consortia have unique metabolic phenotypes in vitro and within the GIT, and that several metabolites were uniquely altered based on microbial composition. For example, in the lower GIT, shikimate and raffinose were utilized by both the OMM12 and SPF microbiota, nicotinate was produced by both microbiotas, and allantoin was consumed only by the SPF and not OMM12 microbiota. Shikimate is metabolized by bacterial and plant cells and contributes to the production of aromatic amino acids and folic acid[24]. Raffinose is a dietary trisaccharide composed of galactose, glucose, and fructose, that is hydrolyzed to galactose and sucrose by the enzyme α-galactosidase. Although α-galactosidase is not found in the GIT of monogastric mammals, including mice and humans, the ability of bacteria to metabolize raffinose has previously been described[25]. In addition, we observed higher levels of nicotinate in the lower GIT of mice colonized with either OMM12 or SPF. Nicotinate can be derived from dietary tryptophan and is an essential precursor for nicotinamide adenine dinucleotide (NAD). Although nicotinate (and other B group

vitamins) is traditionally described as being required via diet, microbial production of these molecules in the lower GIT is becoming increasingly recognized[26]. The biological relevance of bacterial contributions to B vitamins is not well defined, but the clinical prevalence of B vitamin deficiency suggests that bacterial production alone is not sufficient. The abilities of the SPF and OMM12 microbiotas to similarly affect the abundance of molecules such as shikimate, raffinose and nicotinate suggests that these may be well conserved metabolic pathways in bacteria, or that a specific member of the OMM12 consortia is able to normalize to the levels of SPF mice.

Our data also suggest that OMM12 and SPF microbiotas differ in their ability to generate metabolites from the same dietary substrates. For example, cholate was produced in both SPF and OMM12 colonized mice, while taurocholate was only present in SPF mice. Cholate and taurocholate are both primary bile acids and the conjugation of cholate to taurocholate is thought to happen in the liver[27]. The enzymes that allow for deconjugation of bile acids are found across all major phyla in the intestine (reviewed in[27]), and bacterial colonization is known to play a large role in the composition and bioavailability of bile acids[16]. The increased levels of cholate in both OMM12 and SPF mice indicate that the OMM12 consortia is sufficient to increase cholate production but lacks the enzymatic capacity to perform taurine conjugation. This highlights the differential metabolic capacity of the OMM12 and SPF microbiota within the GIT.

Composition-specific effects were not limited to the GIT as several metabolites were differentially abundant in samples from distal sites in SPF versus OMM12 colonized mice, including indoxyl sulfate. Indoxyl sulfate is a tryptophan metabolite that is generally associated with metabolism of indole in the liver. Indole is generated from tryptophanase-dependent catabolism of tryptophan to indole in the small intestine[28] and as such, the increase in indoxyl sulfate in the small intestine of SPF mice is likely due to enterohepatic recirculation. Tryptophan metabolism is a well-characterized mechanism by which bacteria mediate immune responses and physiology. Depending on the enzymatic milieu, tryptophan can be converted to several metabolites via several distinct pathways[10]. Therefore, the SPF-induced increase in indoxyl sulfate is a likely consequence of microbial tryptophanase production, and a subsequent increase in indole. Both indole and indoxyl sulfate can have a wide range of effects on the host. Indole can play important roles in mucosal homeostasis and immune function by activating the aryl hydrocarbon receptor (AhR)[10]. Indoxyl sulfate, however, is associated with cardiac disease, renal toxicity and vascular pathology, and is therefore an important therapeutic target[29]. The lack of indoxyl sulfate in OMM12-colonized mice suggests that none of the 12 bacterial species present in the OMM12 consortia can produce tryptophanase.

In addition to the microbiota, age had a considerable effect on metabolite profiles, particularly at systemic sites. We observed changes in metabolism in weanlings and adult mice, consistent with previous reports showing age-related metabolic changes in sites such as the brain[30] and liver[31]. In our study, microbiota composition shaped how the metabolome was influenced by age, particularly in the lower GIT where the microbial burden is greatest. Several of the compounds that increased with age in the lower GIT of GF mice remained low throughout development in OMM12- or SPF-colonized mice, including ornithine, histidine, uridine, and glutamine. These are likely dietary compounds that would typically be utilized by the microbiota and, as such, accumulate in higher proportions in the GIT of GF adult mice.

Despite their anatomical separation, age-related changes in metabolism at systemic sites did not escape the influences of the microbiota. However, only a few annotated metabolites that were differentially abundant between 3 and 8 weeks of age were microbially altered. One example of these is 4-acetamidobutanoate, which remained relatively constant in the peritoneal fluid and liver of SPF mice throughout development, but substantially increased in GF

mice between weaning and adulthood. OMM12 mice had an intermediate phenotype, with 4-acetamidobutanoate levels increasing only marginally with age. While the biological significance of 4-acetamidobutanoate is unclear, the specific increase in GF mice may point to bacterial utilization of this metabolite in adulthood. Collectively, these data highlight the broad range of metabolic reactions, age-related or otherwise, that are either altered or not present in the intestines and peripheral tissues of GF mice. Moreover, they underscore the large contribution of microbes to metabolism in the mammalian intestine and beyond.

Biological sex impacts many aspects of physiology and immunology in health[32] and has consequences for downstream disease susceptibility[33]. Metabolism is central to all aspects of health and our data highlights differential metabolic responses to microbial colonization depending on sex. For example, in the cecum and colon, we observed an increased in indoleacetate in male but not female mice that were colonized with SPF microbiota. Indoleacetate is a tryptophan metabolite and an activator of the AhR, which has been shown to elicit protective effects in attenuating inflammation in macrophages and cytokine-mediated lipogenesis in hepatocytes in vitro[34]. In addition, within the lower GIT, we observed differential abundances of adenine and inosine in male and female mice when they were colonized. Adenine is a purine derivative that is converted to adenosine or inosine monophosphate, either of which is converted into inosine. Inosine has a wide array of transcriptional activities[35], and has recently been shown to be a microbially-derived immune adjuvant in cancer immunotherapy[12]. We observed that within the GIT, there was sex dependency on how the microbiota affected the abundance of adenine and inosine with an increased abundance of adenine only in female mice and higher levels of inosine in male mice. This suggests differential microbiota-induced purine metabolism in male and female mice, potentially due to differential exposure to sex hormones. Together, our data suggest the microbiota as a sex-specific stimulator of metabolism and highlights the interplay between sex hormones, microbiota and metabolism as an interesting avenue of future research.

As a final note, our study was performed using semi-targeted methods, whereby the metabolites were identified by matching the spectra to a library of known compounds. While this is important to increase confidence in the metabolite identity, it reduces the number of compounds that can be analyzed. Our analysis from 140 metabolites highlights cases of age- and sex-dependent microbial-associated metabolites where we can confidently identify metabolites. However, there were orders of magnitude more metabolites that we were unable to identify. By performing similar analyses with untargeted spectra, we uncovered additional unidentified compounds that were age- and sex- dependent in their response to microbial colonization. As such, our targeted dataset may be a window into a bigger phenomenon of age- and sex- dependency on how the microbiota impacts metabolism.

Together, our data underscore the interconnectedness of host and bacterial metabolism by describing the extensive role that bacteria play in regulating metabolism, while quantifying the impact of age and sex. We found that the levels of certain metabolites are altered by colonization with either a conventional microbiota or a simplified consortium, whereas others were differentially affected by SPF and OMM12, indicating that their abundance was dependent on the presence or absence of certain taxa. We also found cases where microbially-induced changes in metabolism were dependent on the age and sex of the organism, which highlights important considerations for future research studies. Finally, while our study focused on ≈140 compounds that could be annotated with confidence in our assay, analysis of untargeted data shows that many unknown compounds follow similar patterns of age- or sex-dependency in how the microbiota impacts metabolite abundance. As such, our study

provides a template for the consideration of the microbiome, age, and sex in studies of host metabolism and may be representative of a broader phenomenon of age- and sex- dependent effects of the microbiota on metabolism.

## Methods

### Animal handling

All animal experiments were performed in accordance with the guidelines set for by the Canadian Council for Animal Care and all protocols (AC17-0090 and AC17-0011) were approved by the University of Calgary Health Science Animal Care Committee. GF and OMM12-colonized gnotobiotic mice were housed in flexible-film isolators within the International Microbiome Centre (IMC) and SPF mice were maintained in the Mouse Barrier Unit (MBU) at the University of Calgary. GF mice were routinely tested for the absence of bacteria by aerobic and anaerobic culture, gram staining, and vital dye (DNA-dye Sytox green) staining of caecal contents and all mice were routinely screened for the presence of pathogens. All mice were C57BL6/J that were bred in-house under GF or SPF conditions. All animals were fed identical autoclaved diets (LabDiet Autoclavable Rodent Diet, #5K52, Canadian Lab Diets, Leduc County, AB, Canada) ad libitum and maintained at an ambient temperature of ~24 °C, humidity of ~45%, and with 12-hour light-dark cycle.

### Mouse sample collection

Mice were anaesthetized with isoflurane and blood was collected by retro-orbital bleed to serum separation collection tubes (BD, NJ, USA). Urine was collected from the mouse either before it was euthanized, immediately following euthanization or directly from the bladder following euthanization. Mice were euthanized by cervical dislocation. 1.5 mL of ice-cold sterile PBS was injected into the peritoneal cavity, massaged for 20 seconds then removed and collected. The spleen and a segment of left lobe of the liver was collected. The intestine was excised and content from the entire length of the jejunum, ileum, cecum, and colon were collected. For the purposes of these experiments, the first 10 cm of the small intestine was considered duodenum, and the remaining length was split in half with the proximal segment considered jejunum and the distal segment considered ileum. Following collection, blood was centrifuged (10,000 xg, 10 minutes, 4 °C) and serum transferred into a 1.5 mL Eppendorf tube. All collected samples were placed in liquid nitrogen immediately after collection and stored at −80 °C until processing.

### Mouse sample metabolite extraction

For processing liver tissue, spleen, and intestinal content, ~50 mg of tissue or content was added to a pre-weighed 2 mL safe-lock tube containing a steel bead (3 mm, Qiagen, Hilden, Germany). Tubes were reweighed and 5X v/w of ice-cold 50% methanol was added. Samples were homogenized (2 minutes, 30 Hz), beads were removed, and samples stored at −20 °C for 1 hour. Samples were centrifuged (20, 817 xg, 15 minutes, 4 °C), supernatant was recovered and combined 1:4 with 50% methanol to obtain a final dilution of 11:20. For serum and urine, samples were combined 1:1 with 100% methanol and stored and −20 °C for 1 hour. Samples were centrifuged (20, 817 xg, 15 minutes, 4 °C), supernatant was recovered and combined 1:25 with 50% methanol to obtain a final dilution of 11:50. Samples were centrifuged (20, 817 xg, 15 minutes, 4 °C) and supernatants were recovered and stored at −80 °C until further processing.

### Bacterial culture methods and supernatant extraction

Bacteria were grown in 15 mL culture tubes in 3 mL of modified brain heart infusion broth [37 g/L BHI powder, 0.025% Cystiene-HCl.H2O, 0.025% Na2S.9H2O, 1ug/mL Hemin, 0.5ug/mL menadione, 0.025% mucin]. Media was pre-reduced in an anaerobic chamber (Whitley A95

Workstation; 90% N2, 5% H2 & 5% CO2) for 48 hours then bacterial cultures were started from glycerol stocks. 200 μL of glycerol stock was added to each culture tube and cultures were incubated for 6–36 hours until early plateau phase. After incubation, cultures were removed from the anaerobic chamber, and 50uL was collected to extract DNA for 16 S sequencing to confirm accurate identification of bacteria and absence of contamination. The remaining culture was centrifuged (20817 xg, 15 minutes, 4 °C) and the supernatant was collected and combined 1:1 with 100% methanol. Samples were then incubated at − 20 °C for 1 hour. Samples were centrifuged (20,817 xg, 15 minutes, 4 °C), supernatant was recovered and combined 1:4 with 50% methanol to obtain a final dilution of 11:20.

### Liquid chromatography-mass spectrometry

All metabolomics data were collected at the Calgary Metabolomics Research Facility (CMRF) using methods that have been described in detail elsewhere[36,37]. Briefly, samples were centrifuged (20,817 xg, 15 minutes, 4 °C), then 200 μL was transferred to a deep-well 96-well plate (Thermo FisherF) for LC-MS analysis. Data were collected on a Q Exactive™ HF Hybrid Quadrupole-Orbitrap™ Mass Spectrometer (Thermo-Fisher) coupled to a Vanquish™ UHPLC System (Thermo-Fisher). Metabolites were chromatographically separated on Syncronis HILIC UHPLC column (2.1 mm x 100 mm x 1.7um, Thermo-Fisher) at the flow rate of 600uL/min using a binary solvent system (solvent A, 20 mM ammonium formate pH 3.0 in MS grade $H_2O$ and solvent B, MS grade acetonitrile with 0.1% formic acid (%v/v)) and the following gradient: 0–2 mins, 100 %B; 2–7 mins, 100-80 %B; 7–10 mins, 80-5 %B; 10–12 mins, 5% B; 12-13 mins, 5–100 %B; 13–15 mins, 100 %B. The samples injection volume was 2uL. MassM data were acquired in MS1, negative full scan mode at a resolution of 240,000 scanning from 50–750 m/z. Metabolomics data files were processed with ms-mint (version 0.1.8.3; mint.resistancedb.org) Python package for targeted metabolomics and verified using El-Maven (v0.12.0) software package[38,39]. For targeted analysis, metabolites were identified by matching observed m/z signals and chromatographic retention times to those observed from commercial metabolite standards library (MSMLS™ Sigma-Aldrich) containing 639 standards, 397 of which were detectable using our LC-MS and sample preparation methods and 140 of which were observed at detectable levels in the biological samples. For both targeted and untargeted analysis, peaks with a signal intensity <2500 were removed as were peaks that were less than 2 times greater than the average of the blank signal in the region.

### Microbiota composition analysis

DNA extraction and purification from intestinal content was performed using PowerFecal Pro DNA extraction kit (Qiagen, Hilden, Germany). The V4 region of the 16 S rRNA gene was amplified with the following barcoded primer sequences, where "X" indicates an 8-nucleotide barcode (Fwd: AATGATACGGCGACCACCGAGATCTA CAC*XXXXXXXX*TATGGTAATTGTGTGCCAGCMGCCGCGGTAA, Rev: CAAGCAGAAGACGGCATACGAGAT*XXXXXXX*AGTCAGTCAGCCGGA CTACHVGGGTWTCTAAT) using KAPA HiFi polymerase (Roche, Basel, CH) under the following cycling conditions: initial denaturation 98 °C for 2 min, 25 cycles of 98 °C for 30 sec, 55 °C for 30 sec, 72 °C for 20 sec and final elongation at 72 °C for 7 min. NucleoMag® NGS (Macherey-Nagel, Düren, Germany) was used for PCR clean-up and size selection followed by PCR product normalization with the SequalPrep™ Normalization Plate Kit (ThermoFisher, MA, USA) according to the manufacturer's protocols. Individual PCR libraries were pooled, then qualitatively and quantitatively assessed on a High Sensitivity D1000 ScreenTape station (Agilent, CA, USA) and on a Qubit fluorometer (ThermoFisher). 16 S rRNA v4 gene amplicon sequencing was performed using a V2-500 cycle cartridge (Illumina, CA, USA) on the MiSeq platform (Illumina). Miseq data was converted to fastq format and demultiplexed using bcl2fastq software

(Illumina). Sequence processing was performed using the dada2 R package. Sequences with quality score <Q20 were removed, and forward and reverse reads were trimmed to 230 and 210 base pairs, respectively. Sequences were merged, and chimeras were identified and removed. Sequence depth was normalized to 25000 sequences per sample. Taxa were assigned using an in-house database containing the 16 S gene sequences of the OMM12 consortia members. The in-house database was created by taking the 16 S portion of the full-length sequences for the OMM12 members[19] and combining them in fasta format.

### Statistical analysis

Normalized metabolite abundance data are presented as z-scores:

$$\frac{xi - \mu}{\sigma} \tag{1}$$

Where, xi = original metabolite signal, μ = mean of metabolite signal across all samples and σ = standard deviation of metabolite signal across all samples. Fold change calculations were performed as follows:

$$Log2FC = Log2(B) - Log2(A) \tag{2}$$

Where A and B are the mean values for a given metabolite in group A and B, respectively. Where multiple comparisons are drawn from the same dataset, Bonferroni correction was used to adjust p-values for multiple comparisons. Data analysis was performed in the RStudio environment[40] using R programming language[41]. Data manipulation was performed using packages from the tidyverse collection[42]. Multiparametric analyses were performed using functions from the vegan package[43]. Partial $R^2$ analyses were performed using the rsq package[44]. Plots were generated using the ggplot2[45] and pheatmap[46] packages.

### Reporting summary

Further information on research design is available in the Nature Portfolio Reporting Summary linked to this article.

## Data availability

The raw and processed data generated in this study have been deposited in the Metabolomics Workbench database under project ID: PR001468 (https://doi.org/10.21228/M8ZT5R). 16 S amplicon sequencing data have been deposited at NCBI SRA: PRJNA931649(https://www.ncbi.nlm.nih.gov/sra/PRJNA931649). Source data are provided with this paper.

## Code availability

Example code for generating plots is publicly available at the following repository: https://github.com/kirbrown/microbiome-metabolites.git. In addition, the data underlying the manuscript figures can be visualized through a dashboard application available at: https://github.com/mccoy-geuking/microbe_metabolite_app.

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

## Acknowledgements

We thank the staff of the International Microbiome Centre (IMC). The IMC is supported by the Cumming School of Medicine, Western Economic Diversification, and Alberta Economic Development and Trade (AEDT). KB is supported by graduate funding from the Natural Science and Engineering Research Council of Canada (NSERC), Alberta Innovates Health Solutions (AIHS) and the Killam Foundation. IAL is supported by an Alberta Innovates Translational Health Chair and a grant from the Natural Sciences and Engineering Research Council of Canada (NSERC; DG04547). Metabolomics data were acquired at the Calgary Metabolomics Research Facility, which is supported by the International Microbiome Centre and the Canada Foundation for Innovation (CFI-JELF 34986). SW is supported by a 2017 Large Scale Applied Research Program grant from Genome Canada. KDM is supported by the Killam Memorial Chair, University of Calgary. This work was supported by Canadian Institutes of Health Research (CIHR) grant (PJT-165930) to KDM.

## Author contributions

Conceptualization: K.B., I.A.L., K.D.M. Methodology: K.B., S.W., I.A.L., K.D.M. Formal Analysis: K.B., S.W., R.G. Investigation: K.B., M.D., R.G., V.F. Resources: I.A.L., K.D.M. Writing – Original Draft: K.B., C.A.T., I.A.L., K.D.M. Writing – Review & Editing: K.B., C.A.T., S.W., M.D., I.A.L., R.G., K.D.M. Supervision: I.A.L., K.D.M. Funding Acquisition: I.A.L., K.D.M.

## Competing interests

The authors declare no competing interests.
