## [Peer Review File · Nature Communications]

Microbiota alters the metabolome in an age- and sex-dependent manner in miceREVIEWER COMMENTS

Reviewer #1 (Remarks to the Author):

Review of Nat Comm

Microbially metabolic changes across diverse body sites.

These papers are hard to write so kudos to getting it to a submission point. Although a related study such as my own (Quinn, Nature 2020) looked at GF and SPF female mice, there is no such analysis that has been done for female and male mice and not by age. Here the authors assessed the effect of the microbiome on age and sex across the digestive tract, peritoneal fluid, serum, liver, spleen, and urine. These are important fundamental papers. The data is fantastic, and the presentation of the results is encouraging but in a paper such as this, it is important to clearly highlight key messages (currently it is all over the place). For me the main message I got was that there is a sex and age-dependent manner in how the microbiome influences the metabolome throughout the animal. This message could be significantly cleaned up but the data itself is solid and should be published - once the comments are addressed and the presentation is improved. Below are some of my comments and suggestions (and welcome to disagree with me). It is so wonderful to see the selective control they have over the microbial community with the 12-member community.

The title suggests that a much larger body site selection was made. Nor is a map truly described in the paper-although they do have the data to do so. Also the fluids are not body sites and thus suggest to change the title. It would also make sense if age and sex was in the title as this is a key new and exciting contribution to the literature. Perhaps a suggestion: "Microbiota alter the metabolome in a sex and age dependent manner." Although this is-of course- a choice by the authors.

line 70 list the five major prokaryotic phyla.

Abstract, can drop the "high resolution", this is no longer needed. I also suggest to be more specific how many metabolites were targeted (it seems 40-60 or so but cannot get a sense from the manuscript as written at the moment) as the data is not untargeted (if done untargeted >95% of the molecules that would be described would still be unknown and no unknown features are discussed).

Figure 1, IMO, this is the most important data and most important figure. I suggest splitting each time course in M/F and also splitting the OMM12 and SFP from GF as opposed to adjacent to each other and then keeping all the rest of the organization by GI tract the same. This way patterns of age become visible. In addition for the most interesting and representative time trends, you can show them as a scatterplot with significance indicated (plot of time vs levels and separate out sex and three conditions). It is so that the impact of the different conditions can be compared, which is the key. Perhaps even do a PCoA, in addition, of all the measured metabolite values color-coded by each of the conditions and time (perhaps as two plots of the same PCoA, one colored with conditions, other with time).

Figure 2 is not a map. I presume this is an upset plot. Y axis unclear is this percent or number of metabolites or ?. What includes non-systemic sites?

As Figure 2b only addresses the relationships to organ location and not sex or age, Figure 2c 3 a, b and c seems better in the SI. 2A and B get to the new science that is shaping your message.

Figure 4B, it is unclear how the data from individual trains in the 12-member consortium is leveraged to perhaps is connected to the MMVEC results (and should this be done on the 12 member consortium). Perhaps think of it using it the other way that you did MMVEC to find associations and then assessed if such key members could be responsible.

I will stop going through figures as I am not seeing any figs that really pull out the differences in sex or age in relation to body size. You have the data but not yet clearly shown.

The mass spectrometry section has too little information to judge what was done. How many molecules were identified for example? Was only MS1 or also MS2 considered, how about multiple ion forms, etc? If no MS/MS was used then were co-migrations performed with standards? No reference to MAVEN. What identification level are these according to the metabolomics standards initiative? Suggestion. Please improve the description of the metabolomics data collection and annotation.

I did not see a data availability section. I suggest to please add this with accession numbers.

Many reviewers prefer to have figs and fig legends together. I know in the 1990s and before this was used for typesetting as articles were mailed for review but it is really helpful for reviewers on a laptop to have them combined, preferably even in the text where they are supposed to be. Make it as easy for reviewers as possible in my advice, they will be happier.

Final advice/suggestion – and of course is up to the authors as it is their paper. Often less is more. In this case it is likely that the data can be presented in a way to get the key points across in three figures or less. Here is a suggestion for streamlining I can see possible (there are many other ways to do this.) Fig 1. Your general observations and you are close already. Fig 2 your age dependency and this included the MMVEC or similar results what microbes appear to drive metabolite changes fig 3 your sex dependence and again what organisms drive (or associate with sex in relation to organ). All other figs into the SI or omit. The reason for streamlining is because it forces you to focus on the important messages and not speculate all over the place as it is done now and tightening up the message will usually trim such speculation. By not focusing on your messages, it dilutes the impact of the work. In a study like this, one lets the data speak for itself or hypothesize and then performs follow-up experiments to validate (something that was not done hence the suggestion to streamline as it is a beautiful survey on the effect of microbiota.)

Good luck with the resubmission and also welcome to reach out to have a discussion if you like, Pieter

Reviewer #2 (Remarks to the Author):

Brown et al. presents a metabolomic profiling study to better understand how gut microbiota contributes to host metabolism in mice. This is an important study as it has identified microbiota-dependent metabolism that is body site, age, or sex-specific, and these metabolomic profiles can be a helpful resource to the field. Here are several comments for the authors:

Main points:

1. Several recent studies aimed to characterize the impact of gut microbiota on host metabolism. For instance, Quinn et al. (Nature, 2020) surveyed metabolites present in diverse host body sites (e.g., gastrointestinal tract sections, liver, bladder) between germ-free (GF) and specific-pathogen-free (SPF) mice. It will be helpful if the authors can highlight the strength of their metabolomic approach and the key biological discoveries. For example, which classes of bacterial metabolites are detected well by the HILIC negative method used in this paper? For another example, are all metabolites reported in figures and tables at high confidence due to an accompanying in-house library?
2. Brown and colleagues used metabolomics as a foundation for constructing their resource and making biological observations. Thus, it is important to ensure the details of their metabolomics method and analyses are clearly described in the manuscript. Specifically, the HILIC negative method should be described in more detail. On metabolite identification, the authors stated that they used both targeted (relying on in-house chemical standards) and untargeted (based on ms/ms spectral data) analyses. Because the use of in-house standards increases the confidence level of metabolite identification (Schymanski et al, Environ. Sci. Technol, 2014), authors should consider annotating in the figures/tables, whenever applicable, whether the metabolites were identified via targeted or untargeted analyses. The authors should also consider confirming the key metabolites initially identified by untargeted analyses using their matching chemical standards.
3. In several figures, normalized metabolite abundance was used. Can authors describe how they normalized their mass spec raw data? For instance, in Fig 4C, what were media control metabolite levels normalized to (e.g., to the mean value of the four replicates in the media control group)? Including brief explanations in the figure legends or the method section will be very helpful.
4. To enable others to conduct independent analyses on the metabolites identified by targeted and untargeted approaches, the authors should consider releasing supplementary tables in the form of 1) raw ion counts prior to normalization per metabolite, and 2) normalized data per metabolite. I would like to acknowledge that the authors have deposited all unprocessed mass spec raw files at the Metabolomics Workbench.
5. For Fig. 3B, were the Venn diagrams constructed using data from all mice from each colonization state

(including all ages and both sexes)? Would using just the male or female mice increase the number of overlapping metabolites, assuming there may be sex-specific differences?

Minor point:

To indicate the colonization status of each mouse group in the figure legends (e.g., Fig. 3A and 3C), the term of “colonization” may be better than “hygiene.”

Reviewer #3 (Remarks to the Author):

The authors analyzed the metabolomes of germ-free, gnotobiotic, and SPF mice across nine body sites. Subsequently, the authors evaluated the extent by which microbiota, age and sex contributed to variations in the metabolomes. Not surprisingly, the gut microbiota had a huge influence on the metabolome in the gastrointestinal tract (GIT), but also showed significant effects on systemic sites and fluids. However, very few connections were made between changes in metabolites in the gastrointestinal tract and metabolite changes in the systemic sites, and no direct links to specific bacterial species or pathways were deduced or explained in further detail. The explorative work contributes with new hypotheses on metabolites and pathways that could be targeted through gut microbiome alterations.

Main comments

1. It would be really valuable if it was more clear, which metabolites that are changing in abundance due to colonization per say (e.g. taurine and citrulline) and which metabolites that are changing depending on the composition of the microbiota (OMM12 vs SPF, e.g. indoxyl sulfate). I think it would be great if these microbiota-composition-dependent metabolites could be highlighted in figure 3c. Also, I think it would be worth highlighting which metabolites that were not detected in GF-mice. I really think this is what could make the paper stand out compared to the existing papers showing that bile acids, indoles, SCFAs can be manipulated by the gut microbiota (GF vs CONV mice).

2. I am surprised that despite the large effects on metabolites in the GIT, almost none of these affected metabolites changed in serum. Other studies have shown large effects on the plasma metabolome when comparing GF vs CONV mice. Could the authors discuss why so few metabolites were significantly different in serum and liver?

3. The authors show changes in some N-acetyl-amino acids. However, not all of these N-acetyl metabolites are shown in Figure 5. For example, is N-acetylthreonine, N-acetyl-glutamine, N-acetyl-serine and N-acetyltryptophan not presented and discussed. I think this should be better balanced and discussed. Why do only some change? And how do you explain this?

4. The authors culture and grow different bacteria in a rich medium. Subsequently, a number of metabolites that differ between culture supernatants are presented. However, I find it hard to connect

these results (Figure 4c) to the heatmaps (Figure 4d,e) – do any of your in vitro results support your in vivo findings? Furthermore, I see very little overlap between the associations identified between the bacterial species and the metabolites in GIT and systemic sites, respectively. How should this be interpreted? I think the conclusions made from these results currently are rather weak.

5. Are there any evidence suggesting that SCFAs are correlated with levels of these hydroxylated SCFAs? Are you certain that they originate from bacterial fermentation of dietary fibres and proteins? I think the authors should be careful here.

6. I think the authors should be careful with linking differences in mannitol and glucosamine to differences in host energy balance (carbohydrate metabolism).

7. The authors should provide more information on the metabolomics method used since this is really key for the paper. Please provide more details on LC separation and MS data acquisition parameters. Furthermore, were the samples randomized before being analyzed? Were any quality control (QC) samples and/or internal standards analyzed along with the samples in the metabolomics pipeline? If yes, were all metabolites equally reproducible? How did you define quality peaks (Signal-to-noise ratio, minimum intensity, etc)? Which metabolites were analyzed by targeted and untargeted metabolomics, respectively? How were the metabolome (peak areas) normalized across samples?

8. Were the statistical tests adjusted for multiple testing?

Minor comments

Line 44: SCFAs are also produced to some extent by fermentation of proteins.

Line 95: can you give examples of metabolites predominantly regulated by the microbiota across all sites?

Line 201: 3-2-hydroxyphenylpropanoate should be written: 3-(2-Hydroxyphenyl)propanoate

Line 250: Hippuric acid has not been positively associated with insulin regulation, but associated with improved insulin regulation.

POINT-BY-POINT RESPONSE TO REVIEWER COMMENTS

We would like to thank the reviewers for their critical comments and suggestions to improve our manuscript. As a result, we have revised the manuscript considerably and performed new analysis. We provide below a list of changes to the figures, followed by a point-by-point response to each of the reviewers' comments. All reviewers' comments are shown in italics, followed by our response.

List of changes to Figures:

Note: Our original dataset was picked using MAVEN freeware, which has limitations with respect to the number of files that can be processed in the batch. To avoid any batch-to-batch variations in data processing we have updated our analysis in MINT (<https://www.lewisresearchgroup.org/software-tools>), a recently-launched software tool built specifically for large cohort studies. As such, all figures have been remade using a new peak table and slight variation may be present from the original dataset. Below is a list of major changes to the concepts presented in each figure.

Figure 1.

- The metabolites shown in the heatmap in Figure 1A have been changed from the most variable metabolites based on site to the overall most variable metabolites. This has allowed us to visualize microbiota-, age-, and sex- based changes in metabolite abundance along the GIT.
- In Figure 1B, we have selected metabolites from Figure 1A that represent the major classes of metabolites and shown them as scatterplots showing site-, microbiota-, age- and sex- dependency in abundance along the GIT.

Figure 2.

- In Figure 2B, we have included the number of metabolites that were altered based on age-, and sex- at each site in addition to microbiota. We have also changed this to a percentage.
- Original Figure 2C has been removed.
- Updated Figure 2C highlights microbiota-, age- and sex- contributions to individual metabolite concentration by calculating the partial coefficient of determination (R^2) for each metabolite within each site and depicting the partial R^2 as a heatmap. Figure 2C shows the 20 metabolites where the most variation was explained by the 3 factors across all sites. A heatmap showing analysis for all metabolites is presented in Extended Figure 1.

Figure 3.

- Original Figure 3A has been moved to Extended Figure 2C and Extended Figure 2 has been updated to include PCA from all samples sites rather than only the GIT.
- Original Figure 3B has been moved to 3A and expanded to include all sample sites rather than only sites within the GIT.
- New Figure 3B is a biplot generated from the metabolites that were significantly different in either SPF vs. GF and/or OMM12 vs. GF at each site. Extended Figure 4A shows volcano plots comparing SPF vs. GF and OMM12 vs. GF at each site and Extended Figure 4B shows a biplot like Figure 3B but with all metabolites labelled.
- Original Figure 3C has been removed from the manuscript.
- New Figure 3C shows the relative abundance of highlighted metabolites from Figure 3B.
- A data table that forms the base of the biplot is included in supplementary information.

Figure 4.

- Original Figure 4 has been moved to Extended Figure 3. We have also removed original panel E from this figure.

- Extended Figure 5A presents volcano plots of metabolites that were significantly different at each site in 3-week-old vs. 8-week-old mice in GF, OMM12 and SPF conditions.
- New Figure 4A shows a biplot generated from the metabolites that were significantly different in 3-week-old vs 8-week-old mice in GF and/or SPF conditions. Extended Figure 5B shows the same data as Figure 4A but with all metabolites labelled.
- Figure 4B shows the relative abundance of highlighted metabolites from Figure 4A.
- Extended Figure 5C shows the same analysis as Figure 4A and Extended Figure 5B but with fully untargeted peaks where the identity of the compounds is unknown.
- A data table that forms the base of the biplots for both targeted and untargeted analyses are included in supplementary information.

Original Figures 5-8 were removed from the manuscript.

Figure 5.

- Extended Figure 6 shows volcano plots indicating metabolites that were different at each site in male and female mice at 3-, 8-, and 12- weeks of age and under GF, OMM12 and SPF conditions.
- New Figure 5A shows a biplot generated from the metabolites that were significantly different in adult male vs. female mice in GF and/or SPF conditions. Extended Figure 7A shows the same data as Figure 5A but with all metabolites labelled.
- Figure 5B shows the relative abundance of highlighted metabolites from Figure 5A.
- Extended Figure 7B shows the same analysis as Figure 5A and Extended Figure 7AB but with fully untargeted peaks where the identity of the compounds is unknown.
- A data table that forms the base of the biplots for both targeted and untargeted analyses are included in supplementary information.

Point-by-point response to reviewer's comments:

Reviewer #1: *These papers are hard to write so kudos to getting it to a submission point. Although a related study such as my own (Quinn, Nature 2020) looked at GF and SPF female mice, there is no such analysis that has been done for female and male mice and not by age. Here the authors assessed the effect of the microbiome on age and sex across the digestive tract, peritoneal fluid, serum, liver, spleen, and urine. These are important fundamental papers. The data is fantastic, and the presentation of the results is encouraging but in a paper such as this, it is important to clearly highlight key messages (currently it is all over the place). For me the main message I got was that there is a sex and age-dependent manner in how the microbiome influences the metabolome throughout the animal. This message could be significantly cleaned up but the data itself is solid and should be published - once the comments are addressed and the presentation is improved. Below are some of my comments and suggestions (and welcome to disagree with me). It is so wonderful to see the selective control they have over the microbial community with the 12-member community.*

Thank you for your insight. We greatly appreciate your feedback and suggestions and have substantially revised the text and figures following your recommendations.

The title suggests that a much larger body site selection was made. Nor is a map truly described in the paper-although they do have the data to do so. Also the fluids are not body sites and thus suggest to change the title. It would also make sense if age and sex was in the title as this is a key new and exciting contribution to the literature. Perhaps a suggestion: "Microbiota alter the metabolome in a sex and age dependent manner." Although this is-of course- a choice by the authors.

Thank you for this suggestion. We have updated the title and made changes to the manuscript to better reflect the updated title.

Line 70 list the five major prokaryotic phyla.

This has been added.

Abstract, can drop the “high resolution”, this is no longer needed. I also suggest to be more specific how many metabolites were targeted (it seems 40-60 or so but cannot get a sense from the manuscript as written at the moment) as the data is not untargeted (if done untargeted >95% of the molecules that would be described would still be unknown and no unknown features are discussed).

‘High resolution’ has been removed and we have updated the methods sections to better describe the details of the metabolomics. Data presented in the manuscript is from targeted analysis except Extended Figure 5C and Extended Figure 7B (new data), which is untargeted. For targeted analysis, the peaks were compared to an in-house library containing 639 compounds (MSMLS™ Sigma-Aldrich), 397 of which were detectable through our LC-MS methods. Of the 397 detectable metabolites, approximately 140 were detected in our samples.

Figure 1, IMO, this is the most important data and most important figure. I suggest splitting each time course in M/F and also splitting the OMM12 and SFP from GF as opposed to adjacent to each other and then keeping all the rest of the organization by GI tract the same. This way patterns of age become visible.

Thank you for this suggestion. We had selected the most variable metabolites by site to highlight changes along the GIT but have now changed the heatmap to show the most variable metabolites. This has allowed us to better highlight microbiota-, age- and sex- related changes in the metabolome in addition to site changes along the GIT. We have also split GF from OMM12 and SPF as suggested.

In addition, for the most interesting and representative time trends, you can show them as a scatterplot with significance indicated (plot of time vs levels and separate out sex and three conditions). It is so that the impact of the different conditions can be compared, which is the key. Perhaps even do a PCoA, in addition, of all the measured metabolite values color-coded by each of the conditions and time (perhaps as two plots of the same PCoA, one colored with conditions, other with time).

We tried plotting the data different ways to best highlight the impact of the different conditions and decided to highlight a few compounds from each major class of metabolites that were present in the top 40 most variable metabolites. We now show their distribution along the GIT in relation to microbiota, age, and sex (Figure 1B). However, we also present PCoA plots of all the metabolites color coded by site (Extended Fig. 2A), all metabolites in the GIT or other sites (Extended Figure 2B), and split up by individual site (Extended Figure 2C).

Figure 2 is not a map. I presume this is an upset plot. Y axis unclear is this percent or number of metabolites or ?. What includes non-systemic sites? As Figure 2b only addresses the relationships to organ location and not sex or age, Figure 2c 3 a, b and c seems better in the SI. 2A and B get to the new science that is shaping your message.

We agree Fig. 2B and C were not clear. We have now revised Figure 2 to better focus on age-, sex-, and microbial contributions to host metabolism as suggested. Figure. 2A assesses the impact of these three variables in a multiparametric manner (A) while Figure 2B & C assess the contribution of each of these factors to the abundance of individual metabolites. Figure 3A was moved to Extended Figure 2B.

Figure 4B, it is unclear how the data from individual strains in the 12-member consortium is leveraged to perhaps is connected to the MMVEC results (and should this be done on the 12 member consortium). Perhaps think of it using it the other way that you did MMVEC to find associations and then assessed if such key members could be responsible.

We agree that connections between the *in vitro* data and *in vivo* interpretations were not strong as presented. Our intention with the *in vitro* work was to highlight the unique metabolic capacity of individual commensal bacteria. However, when translating to an *in vivo* system – even a simplified microbiota system of 12 members – the potential metabolic interactions that could occur between those species make it very difficult to draw meaningful conclusions and make biological interpretations. Although it would be interesting to be able to tease apart individual microbial contributions to host metabolism using this system, we believe an even more reduced system (mono-colonization or bi-colonization or an *in vitro* co-culture system) would be required. As such, we have moved the bacterial culture data and co-occurrence between metabolites and bacterial abundance to the supplementary material (Extended Figure 3) and adjusted the text to accordingly.

I will stop going through figures as I am not seeing any figs that really pull out the differences in sex or age in relation to body size. You have the data but not yet clearly shown.

We agree with your assessment that we should have better highlighted the differences in sex and age in relation to body site. We have re-analyzed our data and changed multiple figures to better illustrate these differences.

The mass spectrometry section has too little information to judge what was done. How many molecules were identified for example? Was only MS1 or also MS2 considered, how about multiple ion forms, etc? If no MS/MS was used then were co-migrations performed with standards? No reference to MAVEN. What identification level are these according to the metabolomics standards initiative? Suggestion. Please improve the description of the metabolomics data collection and annotation.

Thank you pointing this out. the description of the metabolomics and mass spectrometry has been updated in text (lines: 439-466). The LC-MS methods used in this study have been described in detail elsewhere [1] as have our methods for identifying metabolites [2]. Briefly, metabolomics data are acquired using hydrophilic interaction liquid chromatography (HILIC) on a high-resolution orbitrap mass spectrometer. All metabolites reported here were identified based on high resolution MS1, co-migration of target signals with standards, and MS2 annotation (in cases of ambiguous MS1 assignments) using a commercial library of 639 compounds (MSMLS™ Sigma-Aldrich). References to MAVEN were added as requested.

[1] Groves et al., *Analytical Chemistry* **2022** 94 (25), 8874-8882

[2] Rydzak, et al. *Nat Commun* **13**, 2332 (2022).

I did not see a data availability section. I suggest to please add this with accession numbers.

We have added a data availability section and have uploaded all raw and processed data to Metabolomics Workbench, which is available at the following temporary links. The data will become public upon publication.

Mouse sample data (Study ID: ST002288):

<http://dev.metabolomicsworkbench.org:22222/data/DRCCMetadata.php?Mode=Study&StudyID=ST002288&Access=NusS9188>

Bacterial culture data (Study ID: ST002289):

<http://dev.metabolomicsworkbench.org:2222/data/DRCCMetadata.php?Mode=Study&StudyID=ST002289&Access=TdrA8580>

Many reviewers prefer to have figs and fig legends together. I know in the 1990s and before this was used for typesetting as articles were mailed for review but it is really helpful for reviewers on a laptop to have them combined, preferably even in the text where they are supposed to be. Make it as easy for reviewers as possible in my advice, they will be happier.

Noted, thanks. We have added the Figure legends to each Figure.

Final advice/suggestion – and of course is up to the authors as it is their paper. Often less is more. In this case it is likely that the data can be presented in a way to get the key points across in three figures or less. Here is a suggestion for streamlining I can see possible (there are many other ways to do this.) Fig 1. Your general observations and you are close already. Fig 2 your age dependency and this included the MMCVEC or similar results what microbes appear to drive metabolite changes fig 3 your sex dependence and again what organisms drive (or associate with sex in relation to organ). All other figs into the SI or omit. The reason for stream-lining is because it forces you to focus on the important messages and not speculate all over the place as it is done now and tightening up the message will usually trim such speculation. By not focusing on your messages, it dilutes the impact of the work. In a study like this, one lets the data speak for itself or hypothesize and then performs follow-up experiments to validate (something that was not done hence the suggestion to streamline as it is a beautiful survey on the effect of microbiota.)

We thank the reviewer for this suggestion, which we have carefully considered and incorporated into our manuscript. We have included in Figure 2 some univariate data to complement the multiparametric statistics and to show how individual metabolites are influenced by age, sex, and microbiota. Additionally, we updated Figure 3 to show changes that were driven by SPF or OMM12 microbiota throughout the mouse. In updated Figures 4 & 5, we have highlighted microbe-induced changes in relation to age- and sex. This analysis highlighted a few age- and sex-specific metabolites that we have focused on for discussion. As such, we have removed original Figures 5-8 from the manuscript.

Good luck with the resubmission and also welcome to reach out to have a discussion if you like, Pieter

Thank you for all your comments!

Reviewer #2: *Brown et al. presents a metabolomic profiling study to better understand how gut microbiota contributes to host metabolism in mice. This is an important study as it has identified microbiota-dependent metabolism that is body site, age, or sex-specific, and these metabolomic profiles can be a helpful resource to the field. Here are several comments for the authors:*

Main points:

1. Several recent studies aimed to characterize the impact of gut microbiota on host metabolism. For instance, Quinn et al. (Nature, 2020) surveyed metabolites present in diverse host body sites (e.g., gastrointestinal tract sections, liver, bladder) between germ-free (GF) and specific-pathogen-free (SPF) mice. It will be helpful if the authors can highlight the strength of their metabolomic approach and the key biological discoveries. For example, which classes of bacterial metabolites are detected well by the HILIC negative method used in this paper? For another example, are all metabolites reported in figures and tables at high confidence due to an accompanying in-house library?

Thank you for this comment. In response to the comments from Reviewer 1, we made multiple changes to the figures to better highlight the microbiota, sex, and age differences observed. We have also included greater details about our approach in terms of metabolite detection and identification to the manuscript (lines: 439-466). The LC-MS methods used in this study have been described in detail elsewhere [1] as have our methods for identifying metabolites [2]. Briefly, metabolomics data are acquired using hydrophilic interaction liquid chromatography (HILIC) on a high-resolution orbitrap mass spectrometer. All metabolites reported here were identified based on high resolution MS1, co-migration of target signals with standards, and MS2 annotation (in cases of ambiguous MS1 assignments) using a commercial library of 639 compounds (MSMLS™ Sigma-Aldrich).

Our HILIC methods are calibrated to provide broad coverage of most water-soluble central carbon metabolites with an emphasis on molecules found in extracellular environments. Examples of metabolites covered by these methods include carbohydrates, amino acids and their catabolites and nucleotides and their derivatives. We report on 140 of these metabolites in this manuscript – all of these assignments have been validated using co-elution with analytical standards.

[1] Groves et al., *Analytical Chemistry* **2022** 94(25), 8874-8882

[2] Rydzak, *et al. Nat Commun* **13**, 2332 (2022).

2. Brown and colleagues used metabolomics as a foundation for constructing their resource and making biological observations. Thus, it is important to ensure the details of their metabolomics method and analyses are clearly described in the manuscript. Specifically, the HILIC negative method should be described in more detail. On metabolite identification, the authors stated that they used both targeted (relying on in-house chemical standards) and untargeted (based on ms/ms spectral data) analyses. Because the use of in-house standards increases the confidence level of metabolite identification (Schymanski et al, Environ. Sci. Technol, 2014), authors should consider annotating in the figures/tables, whenever applicable, whether the metabolites were identified via targeted or untargeted analyses. The authors should also consider confirming the key metabolites initially identified by untargeted analyses using their matching chemical standards.

Thank you for this important comment. One important point of clarification: although untargeted and targeted data are reported, all of these data originate from the same high-resolution LC-MS MS1 spectra. Assignments report on the transect of metabolites for which we have assigned by exact mass, co-retention, and ionization relative to commercial metabolites standards. All compounds reported in this manuscript were verified using this method (i.e. no database searching was used).

3. In several figures, normalized metabolite abundance was used. Can authors describe how they normalized their mass spec raw data? For instance, in Fig 4C, what were media control metabolite levels normalized to (e.g., to the mean value of the four replicates in the media control group)? Including brief explanations in the figure legends or the method section will be very helpful.

Normalized data are presented as z scores:

$$\frac{(x_i - \bar{x})}{sd}$$

where x_i is the observed signal for a metabolite observed in a signal sample, \bar{x} is the mean signal for an individual metabolite observed across a biological class, and sd is the standard deviation of metabolite signals observed across the same biological class.

We have updated the methods to better describe the normalization. Note that Fig. 4C is now Extended Fig. 3, and we have included details about this normalization in the figure legend.

4. To enable others to conduct independent analyses on the metabolites identified by targeted and untargeted approaches, the authors should consider releasing supplementary tables in the form of 1) raw ion counts prior to normalization per metabolite, and 2) normalized data per metabolite. I would like to acknowledge that the authors have deposited all unprocessed mass spec raw files at the Metabolomics Workbench.

We have uploaded all raw, processed, and normalized data to Metabolomics Workbench which is available at the following temporary links. This will become public once the embargo is removed.

Mouse sample data (Study ID: ST002288):

<http://dev.metabolomicsworkbench.org:22222/data/DRCCMetadata.php?Mode=Study&StudyID=ST002288&Access=NusS9188>

Bacterial culture data (Study ID: ST002289):

<http://dev.metabolomicsworkbench.org:22222/data/DRCCMetadata.php?Mode=Study&StudyID=ST002289&Access=TdrA8580>

5. For Fig. 3B, were the Venn diagrams constructed using data from all mice from each colonization state (including all ages and both sexes)? Would using just the male or female mice increase the number of overlapping metabolites, assuming there may be sex-specific differences?

We thank the reviewer for this comment. In original Figure 3B, both ages and sexes were combined in the Venn diagram data. Based on this suggestion and suggestions from other reviewers, we have reanalyzed the data to highlight age- and sex- specific ways in which the microbiota alters metabolome (Figures 4 (age) & 5 (sex), Extended Figure 5, 6 & 7). We believe this addresses the reviewer's suggestion to consider sex-specific differences and their contribution to metabolic profiles.

Minor point: To indicate the colonization status of each mouse group in the figure legends (e.g., Fig. 3A and 3C), the term of "colonization" may be better than "hygiene."

This has been updated in text.

Reviewer #3: *The authors analyzed the metabolomes of germ-free, gnotobiotic, and SPF mice across nine body sites. Subsequently, the authors evaluated the extent by which microbiota, age and sex contributed to variations in the metabolomes. Not surprisingly, the gut microbiota had a huge influence on the metabolome in the gastrointestinal tract (GIT), but also showed significant effects on systemic sites and fluids. However, very few connections were made between changes in metabolites in the gastrointestinal tract and metabolite changes in the systemic sites, and no direct links to specific bacterial species or pathways were deduced or explained in further detail. The explorative work contributes with new hypotheses on metabolites and pathways that could be targeted through gut microbiome alterations.*

Main comments

1. It would be really valuable if it was more clear, which metabolites that are changing in abundance due to colonization per say (e.g. taurine and citrulline) and which metabolites that are changing depending on the composition of the microbiota (OMM12 vs SPF, e.g. indoxyl sulfate). I think it would be great if these microbiota-composition-dependent metabolites could be highlighted in figure 3c. Also, I think it would be

worth highlighting which metabolites that were not detected in GF-mice. I really think this is what could make the paper stand out compared to the existing papers showing that bile acids, indoles, SCFAs can be manipulated by the gut microbiota (GF vs CONV mice).

We thank the reviewer for this comment. Based on this suggestion and comments from other reviewers we have restructured our manuscript considerably. We believe that updated Figure 3 and Extended Figure 4 address this specific recommendation to highlight metabolites that are different in GF vs. OMM12 and GF vs. SPF mice.

2. I am surprised that despite the large effects on metabolites in the GIT, almost none of these affected metabolites changed in serum. Other studies have shown large effects on the plasma metabolome when comparing GF vs CONV mice. Could the authors discuss why so few metabolites were significantly different in serum and liver?

We agree that previous studies that analyzed serum in GF and conventionally colonized mice and have found considerable impact of colonization. However, when comparing our results to several other studies we found that the number of serum metabolites altered by colonization were of similar proportion to our study. For example, Wikoff *et al* (2008, PNAS) found that between 6 and 14% of serum metabolites were significantly impacted by microbiota depending on the mode of detection with 6.6% affected in ESI negative mode, which most closely resembles the mode used in our study. In a more recent study, Quinn *et al* (2020, Nature) profiled several sites including the serum and detected approximately 250 unique spectra of which >80% were shared in GF and SPF mice. We observed 5-10% of metabolites are altered by colonization in systemic sites (highlighted in updated Figure 2B) and as such, we believe our study agrees with previously published literature.

3. The authors show changes in some N-acetyl-amino acids. However, not all of these N-acetyl metabolites are shown in Figure 5. For example, is N-acetylthreonine, N-acetyl-glutamine, N-acetyl-serine and N-acetyltryptophan not presented and discussed. I think this should be better balanced and discussed. Why do only some change? And how do you explain this?

This is a good point. Since N-acetylation can occur through host- and bacteria- mediated mechanisms it is possible that N-acetylation of some compounds is dependent on certain bacterial species whereas N-Acetylation of others occurs through host mechanisms. However, through the revision process and helpful suggestions from this reviewer and others, we have shifted the focus of the manuscript away from the individual metabolites identified as differentially abundant at several sites and focused on age- and sex-dependent differences in response to colonization. As such, the previous Fig. 5 has been removed from the manuscript.

4. The authors culture and grow different bacteria in a rich medium. Subsequently, a number of metabolites that differ between culture supernatants are presented. However, I find it hard to connect these results (Figure 4c) to the heatmaps (Figure 4d,e) – do any of your *in vitro* results support your *in vivo* findings? Furthermore, I see very little overlap between the associations identified between the bacterial species and the metabolites in GIT and systemic sites, respectively. How should this be interpreted? I think the conclusions made from these results currently are rather weak.

We agree that connections between the *in vitro* data and *in vivo* interpretations were not strong as presented. Our intention with the *in vitro* work was to highlight the unique metabolic capacity of individual commensal bacteria. However, when translating into an *in vivo* system – even a simplified microbiota system of 12 members – the potential metabolic interactions that could occur between those species made it very difficult to draw meaningful conclusions and make biological interpretations. Although it would be interesting to be

able to tease apart individual microbial contributions to host metabolism using this system, we believe an even more reduced system (mono-colonization or bi-colonization or an *in vitro* co-culture system) would be required. Unfortunately, we believe this is out of scope of the current manuscript. As such, we have moved the bacterial culture data and co-occurrence between metabolites and bacterial abundance in the supplementary material (Extended Figure 3) and have removed focus from this aspect of our manuscript.

5. Are there any evidence suggesting that SCFAs are correlated with levels of these hydroxylated SCFAs? Are you certain that they originate from bacterial fermentation of dietary fibres and proteins? I think the authors should be careful here.

We thank the author for this comment as it is true that we could not be sure of the source of these metabolites. With the restructuring of the manuscript, there is no focus on these metabolites and as such we no longer include statements about their origin.

6. I think the authors should be careful with linking differences in mannitol and glucosamine to differences in host energy balance (carbohydrate metabolism).

Similar to the above comment, we have now restructured the manuscript and no longer make connections between these metabolites and host energy balance.

7. The authors should provide more information on the metabolomics method used since this is really key for the paper. Please provide more details on LC separation and MS data acquisition parameters. Furthermore, were the samples randomized before being analyzed? Were any quality control (QC) samples and/or internal standards analyzed a long with the samples in the metabolomics pipeline? If yes, were all metabolites equally reproducible? How did you define quality peaks (Signal-to-noise ratio, minimum intensity, etc)? Which metabolites were analyzed by targeted and untargeted metabolomics, respectively? How were the metabolome (peak areas) normalized across samples?

The LC-MS methods used in this study have been described in detail elsewhere [1] as have our methods for identifying metabolites [2]. Briefly, metabolomics data are acquired using hydrophilic interaction liquid chromatography (HILIC) on a high-resolution orbitrap mass spectrometer. All metabolites reported here were identified based on high resolution MS1, co-migration of target signals with standards, and MS2 annotation (in cases of ambiguous MS1 assignments) using a commercial library of 639 compounds (MSMLS™ Sigma-Aldrich).

Regarding targeted and untargeted metabolomics – these terms are somewhat misleading all data originate from the same high resolution LC-MS MS1 datasets, what differentiates these datasets is that we have curated a selection of signals using our established metabolite assignment methods. We report on 140 of these metabolites in this manuscript – all of these assignments have been validated using co-elution with analytical standards.

We have included details about our approach in terms of metabolite detection and identification to the manuscript (lines: 439-466).

[1] Groves et al., *Analytical Chemistry* **2022** 94 (25), 8874-8882

[2] Rydzak, et al. *Nat Commun* **13**, 2332 (2022).

8. Were the statistical tests adjusted for multiple testing?

Yes, in all cases where univariate statistics were presented, we adjusted alpha thresholds to account for multiple test penalties via the Bonferroni correction method. This has been clarified in methods.

Minor comments

Line 44: SCFAs are also produced to some extent by fermentation of proteins.

Thanks, this has been updated in text.

Line 95: can you give examples of metabolites predominantly regulated by the microbiota across all sites?

With the shift in focus of the manuscript, we no longer focus on metabolites that are regulated across all sites.

Line 201: 3-2-hydroxyphenylpropanoate should be written: 3-(2-Hydroxyphenyl)propanoate.

This has been updated in text.

Line 250: Hippuric acid has not been positively associated with insulin regulation, but associated with improved insulin regulation.

With updates to the manuscript, we no longer make this reference.

REVIEWERS' COMMENTS

Reviewer #1 (Remarks to the Author):

The authors responded very well and took the comments by reviewers into consideration and really drove home the key points with respect to the impact of the microbiome (and their nicely controlled 12-bacterial community), age, and sex. This was hidden last time and this was my main concern. This is now very well addressed. I really like the quadrant plots when comparing conditions, I may borrow this concept in my future papers as well. It is a very simple approach to getting the message across. In my opinion, every figure is now very easy to interpret with respect to the microbial community, age, and sex dependencies. Some interesting surprises wrt specific metabolites-in particular some of the metabolites generally attributed to the microbiome that are sex-specific. I can't wait to see how other labs may use this information to understand sex-specific microbiome phenotypes and health conditions. It is a nice contribution to the literature. I was hoping to directly look at the data but I could not find the data deposition when I went to metabolomics WB and searched the accessions (I normally quickly look at some of the results at the raw data level). I do know some repositories are behind in their curation to make it public. I will make the recommendation to accept but also tell the editor to accept upon ensuring that the data is public.

Best, Pieter

Reviewer #2 (Remarks to the Author):

The authors have addressed my concerns.

Reviewer #3 (Remarks to the Author):

The authors have really improved the clarity of their work. Well done! The presentations of age, sex, and microbiota-dependent metabolome differences and the comparison of effect sizes are much better now. I think the work makes an important contribution to the field.

Few minor comments, which the authors may consider discussing.

I think it is worth mentioning that indoxyl-sulphate (and also hippurate) is predominantly measured in the small intestine, which may suggest that indole (and hippurate) is formed in the small intestine where it is absorbed to a large extent (and maybe recirculated via the enterohepatic circulation?). In agreement, you find the compound in serum, liver, spleen and urine. I think this is key information to the community, since proteolytic fermentation (indole formation) is often considered to be related to colonic fermentation.

I do not understand the sex-specific effects on indoleacetate, which can either derive from the feed (plant-material) or the gut microbes (as you mention). How would you explain this? Are you certain that it is indeed indoleacetic acid you identified? I would have expected this metabolite to either be increased in SPF/OMM12 mice compared to GF mice (if it was derived from microbes) or to be similar in all mice (if it was derived from the feed).

Figure 1a – could you repeat the info on GF, OMM12, SPF and weeks (3, 8, 12) below all compartments (like you do for jejunum)?

Figure 1b is a little difficult to look at. You may consider moving the panel to an extended figure.

Line 142: Please correct sentence when proof-reading :-)

Line 193: Please correct sentence when proof-reading

/Henrik Roager

We would like to thank the reviewers for their comments throughout the review process and suggestions to improve our manuscript. We have addressed the final reviewer comments in a point-by-point format below and have addressed the editorial comments in the attached author checklist document. All data has been made publicly available.

Response to reviewers. Our responses are in *italics* below.

REVIEWERS' COMMENTS

Reviewer #1 (Remarks to the Author):

The authors responded very well and took the comments by reviewers into consideration and really drove home the key points with respect to the impact of the microbiome (and their nicely controlled 12-bacterial community), age, and sex. This was hidden last time and this was my main concern. This is now very well addressed. I really like the quadrant plots when comparing conditions, I may borrow this concept in my future papers as well. It is a very simple approach to getting the message across. In my opinion, every figure is now very easy to interpret with respect to the microbial community, age, and sex dependencies. Some interesting surprises wrt specific metabolites-in particular some of the metabolites generally attributed to the microbiome that are sex-specific. I can't wait to see how other labs may use this information to understand sex-specific microbiome phenotypes and health conditions. It is a nice contribution to the literature. I was hoping to directly look at the data but I could not find the data deposition when I went to metabolomics WB and searched the accessions (I normally quickly look at some of the results at the raw data level). I do know some repositories are behind in their curation to make it public. I will make the recommendation to accept but also tell the editor to accept upon ensuring that the data is public.

Best, Pieter

Thank you very much for your comments. Your comments were so helpful to revise our manuscript to ensure the key points were clear. We apologise the raw data was not accessible, it should have been. We are making sure the data is now public.

Reviewer #2 (Remarks to the Author):

The authors have addressed my concerns.

Thank you for your input.

Reviewer #3 (Remarks to the Author):

The authors have really improved the clarity of their work. Well done! The presentations of age, sex, and microbiota-dependent metabolome differences and the comparison of effect sizes are much better now. I think the work makes an important contribution to the field.

Thank you very much for your comments.

Few minor comments, which the authors may consider discussing.

I think it is worth mentioning that indoxyl-sulphate (and also hippurate) is predominantly measured in the small intestine, which may suggest that indole (and hippurate) is formed in the small intestine where it is absorbed to a large extent (and maybe recirculated via the enterohepatic circulation?). In agreement, you find the compound in serum, liver, spleen and urine. I think this is key information to the community, since proteolytic fermentation (indole formation) is often considered to be related to colonic fermentation.

It is true that we find indoxyl sulfate (and hippurate) in the small intestine, serum, liver, spleen and urine. Indoxyl sulfate is generated in the liver downstream of indole generation from tryptophan in the intestine. We cannot say precisely where the indole generation occurs, however, since indoxyl sulfate is generated in the liver, the increase in indoxyl sulfate in the small intestine is likely due to hepatic recirculation. We have added a sentence describing this in the text (line 273-4).

I do not understand the sex-specific effects on indoleacetate, which can either derive from the feed (plant-material) or the gut microbes (as you mention). How would you explain this? Are you certain that it is indeed indoleacetic acid you identified? I would have expected this metabolite to either be increased in SPF/OMM12 mice compared to GF mice (if it was derived from microbes) or to be similar in all mice (if it was derived from the feed).

We are reasonably confident that it is indoleacetic acid that we identified. All metabolite assignments, including this indoleacetate example, are supported by 1) exact m/z matching to the assigned compound, and 2) co-elution of the observed signal with a metabolite standard used as a reference compound. All standard reference materials used in this project were purchased via the MS-MSMLS library and exact m/z matching was defined as signals with < 5 ppm difference in observed versus expected exact masses based on the molecular formula. Co-elution was defined as <2 second difference in retention times for the target signal and the reference metabolite standard.

Our interpretation would be that since OMM12 and female SPF levels are similar to GF, this baseline level is likely to be diet-derived. The induction in male SPF mice is dependent on bacteria (absent on GF) and therefore there is something about colonization with SPF microbiota (but not OMM12) in males but not females that causes increased levels of this metabolite. The precise mechanism of how this occurs is beyond the scope of this manuscript but could be related to differences in sex hormones driving changes in the microbiota between males and females. Sex hormones can influence host physiology, microbiota, and/or availability of metabolites for catabolism. This is one example of a metabolite that is differentially affected in male and female mice by colonization that we chose to focus on, but several more metabolites that we did not identify were also affected by colonization (Supplementary Figure 7B).

Figure 1a – could you repeat the info on GF, OMM12, SPF and weeks (3, 8, 12) below all compartments (like you do for jejunum)?

The figure has been updated as suggested.

Figure 1b is a little difficult to look at. You may consider moving the panel to an extended figure.
We appreciate that this figure may be a little difficult to look at but have decided to keep this in the main figures since we only have 5 main figures and we feel the message will be lost if moved to supplementary.

Line 142: Please correct sentence when proof-reading :-)

Thanks, this has been corrected

Line 193: Please correct sentence when proof-reading

Thanks, this has been corrected

/Henrik Roager